# Antioxidant Effect and Sensory Evaluation of Yogurt Supplemented with Hydroponic Ginseng Root Extract

**DOI:** 10.3390/foods10030639

**Published:** 2021-03-17

**Authors:** Hyun Sook Lee, Myung Wook Song, Kee-Tae Kim, Wan-Soo Hong, Hyun-Dong Paik

**Affiliations:** 1Department of Foodservice Management and Nutrition, Sangmyung University, Seoul 51767, Korea; leehs9292@smu.ac.kr (H.S.L.); wshong@smu.ac.kr (W.-S.H.); 2Department of Food Science and Biotechnology of Animal Resources, Konkuk University, Seoul 05029, Korea; carebie251@hanmail.net (M.W.S.); richard44@hanmail.net (K.-T.K.)

**Keywords:** yogurt, hydroponic ginseng, antioxidant effect, sensory evaluation, dairy product

## Abstract

Hydroponic ginseng (HG) is cultivated using only nutrients and water under constant environmental conditions and is more beneficial than soil-cultured ginseng (SG). This study aimed to determine the physicochemical properties, antioxidant activity, and sensory properties of HG-supplemented yogurt to develop high-value yogurt. HG (0.1%, 0.5%, and 1.0%) was added to yogurt formulations and fermented with a 0.1% starter. Antioxidant activities were determined using the 2,2-diphenyl-1-picryl-hydrazyl (DPPH), 2-azinobis-(3-ethylbenzothiazoline-6-sulfonic acid) diammonium salt, reducing power, and ferric reducing antioxidant power assays. Semi-trained panelists performed a quantitative descriptive analysis for sensory evaluation. The number of starter cells increased more rapidly in ginseng extract-fortified yogurt than in the control group, shortening fermentation time. Regarding antioxidant assays, all HG extract-fortified yogurts showed higher antioxidant activity than the control group. In particular, the HG (0.5%) group showed better results than the SG group in the DPPH and reducing power assays, although the difference was not significant. The sensory scores of color, flavor, texture, taste, and overall acceptance of 0.5% HG-supplemented yogurt did not differ significantly from those of non-supplemented yogurt (control). This suggests that HG can be used in high-value dairy products as a supplement with bioactive properties for health in the food industry.

## 1. Introduction

Recently, many consumers have become interested in their health and well-being. Accordingly, high-value food products are being developed actively, and these products are increasingly being used as biofunctional foods [1]. In particular, dairy food companies have researched new products as functional foods with additional biofunctional activities to prevent various diseases and maintain health as well as consumers’ acceptability.

In the human body, oxidative stress occurs owing to reactive oxygen species that are produced aerobically as waste products of metabolism [2,3]. Reactive oxygen species are highly toxic, causing various diseases, and they must be eliminated as quickly as possible by antioxidants through various cellular mechanisms [4].

The bioactivity of dairy products for health has been studied extensively and, in particular, many studies have reported that cheese has antioxidant effects because of the phenolic compounds in dairy products [5,6]. Some studies have reported that the antioxidant benefits are owing to the complex reaction between the milk proteins and phenolic compounds [7]. However, this antioxidant effect may be relatively low because of the low concentrations of polyphenolic compounds in dairy products.

Supplementation of a variety of dairy products with natural ingredients, including herbs, has been studied extensively to increase bioactivity for human health [8]. However, there is still no research on the effects of yogurt supplemented with hydroponic ginseng root extract as a new dairy product.

Ginseng root has been used as a traditional medicinal material in Northeast Asian countries, including Korea. Ginseng root extracts contain many kinds of ginsenosides as major active ingredients and have been known to act against diseases, such as cancer, or to improve human health [9]. However, it is difficult to commercialize these products as the cultivation time in soil is long (a minimum of 4 to 6 years) [10], and there are many potential obstacles to harvest, including pathogens from soil, root rot, the accumulation of residual pesticides, and contamination of heavy metals [11,12]. Recently, interest in hydroponic ginseng has attracted increasing attention as an alternative, as in hydroponics, ginseng is cultivated in mineral nutrient solutions without pesticides. Additionally, the productivity of hydroponic ginseng per unit area is much higher than that with soil cultivation [12]. It was reported that the management cost for a hydroponic cultivation system is approximately 36,000 $/acre, whereas that for soil-cultivation system is approximately 282,000 $/acre [13]. In addition, the control of cultivation in hydroponic systems is easier than in conventional soil farming. Furthermore, unlike soil cultivation in which only the root parts are consumed, all parts of the hydroponic ginseng (leaves, stems, and fruits) can be used commercially. However, although hydroponic ginseng has many merits compared to soil-cultivated ginseng, there are few reports on the biofunctional effects of hydroponic ginseng on health compared to those of traditionally soil-cultivated ginseng.

Recently, there have been several reports on the effects of ginseng extract on dairy products, such as cheese, as high-value products with health-functional activity [14]; however, the application of hydroponic ginseng extract to yogurt as a representative dairy product remains unclear. Therefore, this study aimed to analyze the physiochemical properties and antioxidant effects of yogurt supplemented with hydroponic ginseng extract in vitro, and to evaluate sensory properties such as color and texture.

## 2. Materials and Methods

### 2.1. Materials

Whole cow milk and skim milk powder were acquired from Seoul Milk Co. (Seoul, Korea) and a commercial starter (ABT-B comprising *Lactobacillus acidophilus*, *Bifidobacterium longum*, and *Streptococcus thermophilus*) was obtained from Samik Dairy & Food Co. (Seoul, Korea). Soil-cultured ginseng cultured for six years (SG; Sangdo-Insamsa Co., Punggi, Korea) and hydroponic ginseng cultured for two years (HG; Chungjung-Saessacksam Co., Gwangju, Korea) were purchased from a local market. The other chemicals were purchased from Sigma-Aldrich (St. Louis, MO, USA). Ginseng extract was obtained from Hwain Korea Co. (Seoul, South Korea). Man-Rogosa-Sharpe agar was obtained from Difco Laboratories (Detroit, MI, USA).

### 2.2. Ginseng Extract Preparation

Only the root parts of both SG and HG were used, and each ginseng root was rinsed, cut into small pieces, dried at 60 °C for 24 h, and then milled and separated using a 100-mesh screen. Each sample of ginseng powder (25 g) was extracted with 500 mL of 70% ethanol in a reflux system at 70 °C for 6 h, and the extraction was performed in triplicate. Whole extracts were filtered using Whatman filter paper No. 2 and concentrated using a rotary evaporator (N-1000V, EYELA, Tokyo, Japan) system at 60 °C. The concentrates were lyophilized and then stored anaerobically at 4 °C until use.

### 2.3. Yogurt Preparation

Milk (400 mL) was supplemented with skim milk powder (11.42 g) to yield 11% in solid matter and ginseng extract of different concentrations (C: control yogurt without ginseng extract, SG: 1.0% soil-cultured ginseng extract, HG 0.5: 0.5% hydroponic ginseng extract, and HG 1.0: 1.0% hydroponic ginseng extract) was added and then pasteurized at 90 °C for 10 min. Pasteurized milk was cooled to 25 °C and inoculated with a commercial starter at a final concentration of 0.1%. The milk mixtures were incubated at 40 °C until reaching pH 4.5, and were then stored in a refrigerator at 4 °C at least 24 h for ripening.

### 2.4. Physicochemical Properties

Physicochemical properties such as protein, fat, total solid, ash, titratable acidity, and lactose content of yogurt were evaluated according to the guidelines described by the Association of Official Analytical Chemists (AOAC, 2000) and with MilkoScan™ Minor (Type 78110, FOSS, Hillerød, Denmark) as per the manufacturer’s instructions. Color values were evaluated using a colorimeter (CR-400, Konica Minolta, Tokyo, Japan). The pH value was measured using a pH meter (pH 7110, WTW, Weilheim, Germany) and viable cell counts were measured by plate counting on bromocresol purple medium (MB Cell, Seoul, Korea) every hour.

### 2.5. Antioxidant Activity

#### 2.5.1. Sample Preparation

Antioxidant activity was measured using the supernatant of each yogurt. The ginseng extracts were diluted to 10, 5, and 2.5 mg/mL, and the prepared yogurts were centrifuged (8000× *g*, 4 °C, 15 min) to obtain the supernatant and filtered through a 0.45 μm membrane filter (Advantec, Tokyo, Japan).

#### 2.5.2. Radical Scavenging Activity

Two types of radical scavenging assays (2,2-diphenyl-1-picryl-hydrazyl (DPPH) radical scavenging assay and 2-azinobis-(3-ethylbenzothiazoline-6-sulfonic acid) diammonium salt (ABTS) radical scavenging assay) were performed to estimate the radical scavenging activity of the ginseng extract and yogurt according to the method described by Yu et al. [15] with minor modifications. Each sample and 0.4 mM DPPH solution were mixed in a 1:5 ratio, and then the mixtures were kept at 25 °C for 30 min in dark conditions. The absorbance was measured at 517 nm. ABTS solution was prepared by mixing ABTS (14 mM) and potassium persulfate (5 mM) in 0.1 M potassium phosphate buffer (pH 7.4), and then reacted for 16 h at 25 °C. The reaction mixture was diluted until the absorbance met 0.7 ± 0.05 at 734 nm. Fifty microliters of each sample was mixed with 950 μL of diluted ABTS solution and reacted for 15 min in the dark. After the reaction, the absorbance was measured at 734 nm.

Radical scavenging activity of both assays was measured using the following formula:(1)$$\mathrm{Radical}\;\mathrm{scavenging}\;\mathrm{activity}\;\left(\%\right)=\left(1-\frac{A_{s}}{A_{c}}\right)\;\times 100$$
where *A_s_* and *A_c_* represent the absorbance of the sample and control, respectively.

#### 2.5.3. Reducing Power Assay

The reducing power assay was performed using the method described by Jang and Koh [16] with minor modifications. Fifty microliters of yogurt supernatant was mixed with 250 μL of 0.2 M sodium phosphate buffer (pH 6.6), and 250 μL of 1% potassium ferricyanide, and incubated at 50 °C for 20 min. Subsequently, 250 μL of 10% trichloroacetic acid was added, and 500 μL of the solution was mixed with 400 μL of distilled water and 100 μL of 0.1% ferric chloride. After incubation at 25 °C for 30 min, the absorbance was measured at 700 nm. Reducing power was evaluated using the standard curve of L-cysteine.

#### 2.5.4. Ferric Reducing Antioxidant Power (FRAP) Assay

The FRAP assay was evaluated using the method described by Eom et al. [17] with minor modifications. To prepare the FRAP solution, 300 mM acetate buffer (pH 3.6), 10 mM 2,4,6-tri[2-pyridyl]-s-triazine dissolved in 40 mM HCl, and 20 mM ferric chloride was mixed in a 10:1:1 ratio and incubated at 50 °C for 15 min. Fifty microliters of sample was reacted with 950 μL of FRAP solution and incubated at 25 °C for 30 min. Then, absorbance was measured at 593 nm, and FRAP activity was evaluated using a ferrous sulfate standard curve.

### 2.6. Sensory Analysis

Produced each yogurt was chilled at 4 °C at least 24 h for ripening, and then yogurt samples were delivered to another institution for sensory evaluation. Consumer sensory analysis was performed by 22 semi-trained panelists (12 female and 10 male, 26 to 30 years old), and the protocol was approved by the Institutional Review Board (approval number: IRB-SMU-C-2020-4-004, Korea). Quantitative descriptive analysis was performed to evaluate the differences in the sensory characteristics of ginseng-supplemented yogurt samples [18,19]. Water and plain bread were provided between samples as a palette cleanser and yogurt without the ginseng extract served as the reference standard.

### 2.7. Statistical Analysis

The data are expressed as mean ± standard deviation. For statistical comparisons, the results were subjected to one-way analysis of variance (*p* < 0.05) and the Duncan’s multiple-range test using IBM SPSS 22 software (IBM, Armonk, NY, USA).

## 3. Results

### 3.1. Antioxidant Activity of Ethanol-Extracted Ginseng Extracts

DPPH and ABTS radical scavenging activity assays of SG and HG extracts were conducted in this study, and the results are presented in Figure 1. Each ginseng extract was diluted to three different concentrations (2.5, 5.0, and 10 mg/mL). All sample-treated groups showed dose-dependent effects at all concentrations in both radical scavenging assays. Furthermore, the HG extract consistently showed higher antioxidant activity than the SG extract. In both DPPH and ABTS assays, HG averagely represented 3.18- and 1.30-fold higher activity levels, respectively.

### 3.2. Physicochemical and Microbial Properties of Yogurt

The pH, titratable acidity, and viable cell count were measured during the yogurt fermentation period. The results of pH change and titratable acidity are represented in Figure 2A,B, respectively. The pH value of the yogurt samples decreased from 6.5 to 4.5 over a period of 5 h, and during this time, titratable acidity increased from 0.25 to 1.23. These results imply that changes in pH and titratable acidity are correlated with each other. Ginseng extract-fortified yogurts changed slightly more than control yogurt, and this phenomenon could be caused by the innate acidity of ginseng. The number of viable cells is shown in Figure 2C. These results showed a similar pattern between yogurt samples to those of the previous results. The number of viable cells increased more rapidly in ginseng extract-fortified yogurts than in the control yogurt. The addition of ginseng extracts would provide some nutritional components that aid the growth of lactic acid bacteria.

The fat, protein, lactose, total solid, and ash content and light analysis results of the yogurt samples are presented in Table 1. The yogurt without ginseng extract (control) comprised 3.32% fat, 4.15% protein, 7.28% lactose, 14.71% total solid, and 0.76% ash. Among these factors, protein, total solid, and ash content were affected by the addition of ginseng extract. Protein, total solid, and ash content of ginseng extract-supplemented yogurt showed a dose-dependent increasing pattern with significant differences. Although the concentration of HG 0.5 was twofold lower than that of SG, it contained slightly more protein and ash. Total solid content also showed a dose-dependent increasing pattern. However, the fat and lactose content did not show significant differences between all types of yogurt samples. The color was analyzed using the three parameters L*, a*, and b*, indicating dark to light, green to red, and blue to yellow, respectively. Supplementation of ginseng extract influenced all color parameters in a dose-dependent manner. The L* value decreased from 93.96 to 90.99, and those of a* and b* increased from 2.92 and 2.30 to 5.91 to 11.11, respectively. Indeed, a light and white appearance was observed in the control group, whereas ginseng extract-fortified yogurts were light brown in color. Furthermore, HG-supplemented yogurt was shown to have a relatively more dark and yellow appearance than the SG-supplemented yogurt consistent with the results presented in Table 1.

### 3.3. Antioxidant Results of Ginseng Extract-Fortified Yogurts

To evaluate the antioxidant activity of ginseng extract-fortified yogurt, the supernatants of each yogurt were obtained through centrifugation and filtration. Results of the four antioxidant assays, including DPPH and ABTS radical scavenging assays, reducing power, and FRAP assays, are shown in Figure 3. In all the assays, all ginseng extract-fortified yogurts showed higher antioxidant activity than the control yogurt, regardless of the type of added ginseng. Furthermore, among the ginseng extract-fortified yogurts, HG 1.0 showed the highest antioxidant activity. Although no significant differences were observed between the DPPH and reducing power assays, HG 0.5 presented higher results than SG. These results were consistent with the results of the antioxidant assay for ginseng extract presented in Figure 1.

### 3.4. Sensory Evaluation

The sensory attributes of yogurt supplemented with different concentrations (0.5% or 1.0%) of the different hydroponic ginseng extracts are presented in Table 2. The acceptability of color was increased with the addition of hydroponic root extract below a concentration of 0.5% but was lowered at concentrations above 1.0%. Regarding texture and flavor, it appeared that the scores of both these parameters were not affected at a concentration lower than 0.5%. Flavor properties such as bitterness and ginseng odor significantly increased with increasing concentration of the ginseng extract (*p* < 0.05). In addition, the score for the aftertaste increased with the concentration of ginseng extract. In terms of total quality, there was no significant difference between the non-supplemented control and the samples with less than 0.5% added ginseng extract in yogurt (*p* > 0.05), however, acceptability as a yogurt product was negatively affected at a concentration above 1.0%.

## 4. Discussion

Ginseng (*Panax ginseng* Meyer, Asian ginseng) has been widely consumed as an herbal medicine because of its efficacy as an anticancer-, antioxidant-, anti-inflammatory, and anti-obesity agent [20]. The source of these functional merits is related to various compositions of ginsenoside content as well as phenolic and flavonoid compounds [21]. *Panax* plants are cultivated in approximately 35 countries globally, and the chemical composition of ginseng varies vastly owing to climate, geographical effects, and extraction process. For example, *P. notoginseng* contains more kinds of ginsenosides than *P. quinquefolius* and *P. ginseng* [22]. In this study, the roots of soil-cultured ginseng and hydroponic ginseng were investigated. Hydroponic systems have advantages not only for facilitating productivity but also for controlling environmental factors. This system enables the acceleration of growth and increases the functional compounds of ginseng through the regulation of nutritional supply, light intensity, and temperature [23].

Indeed, the HG extract showed higher antioxidant activity than the SG extract (Figure 1). The radical scavenging activity of the HG extract was 3.18- and 1.30-fold higher than that of the SG extract in the DPPH and ABTS radical scavenging assays, respectively. Therefore, HG extract could be regarded as having nearly a twofold higher functional potential. The addition of ginseng or ginseng marc extracts to yogurt has been reported [18,24]. In these studies, ginseng extract showed a positive effect on the viability of lactic acid bacteria and sensory properties. A high concentration (1.5–2.0%) of the extract weakly inhibited the growth of lactic acid bacteria and increased the bitterness of yogurt. According to these previous studies, 1% of ginseng extract was considered ideal for application in yogurt. In our study, with the addition of 0.5% concentration of ginseng extract to yogurt, the HG extract had antioxidant ability roughly twice that of the SG extract.

The physicochemical properties of ginseng-supplemented yogurt have been studied previously [18,24]. Among various parameters, protein, total solids, and ash were influenced by the addition of ginseng extract. Ginseng generally comprises polysaccharides, amino acids, saponins, volatile oils, and polyacetylenes. Several organic acids, non-saponin water-soluble glycosides, salicylic acid amine, and alkaloids have been identified. In addition to these components, some trace elements, such as vitamins, flavonoids, and enzymes, have also been found in ginseng [25]. Therefore, the physicochemical properties could have been affected by the composition of ginseng in the fortified yogurts. The total solid content also increased with the amount of supplemented ginseng extract. Moreover, these nutritional components might be responsible for the increased growth rate of lactic acid bacteria. Because carbohydrates are major ingredients of ginseng, it can provide a nutritional source for inducing the growth or activity of microorganisms in yogurt. The different colors of yogurt were caused by the native color of ginseng. The exterior appearance of ginseng is dark yellow or light brown, and its interior is light yellow. Concentrated ginseng extract is also dark brown in color; therefore, ginseng extract-fortified yogurts were unaffectedly light brown [18]. As the concentration of ginseng extract increased, lightness (L*) and yellowness (b*) also increased. Although the HG 0.5-fortified yogurt contained a lower ginseng concentration than SG yogurt, it had a higher b* value because of its inherent yellowness. Regarding the a* value, which indicates greenness and redness, there were significant differences between the yogurt samples. In addition, it was difficult to classify any greenness or redness color with the naked eye. Thus, the added-ginseng extracts were considered to have mainly influenced the lightness and yellowness of ginseng yogurts.

In this study, four antioxidant assays were performed on ginseng extract-fortified yogurts. All yogurt samples tested showed antioxidant activity. Among the ginseng yogurts, HG 1.0 in particular showed the highest antioxidant activities in all assays, and HG 0.5 showed the second highest results, similar to those of SG yogurt. Casein and some milk proteins can provide several bioactive peptides through proteolytic degradation, which determined antioxidant properties [26]. Several studies have reported that the antioxidant activity of yogurts decreased until the seventh week, and then slightly increased again after the eighth week during a long-term storage period [27]. This phenomenon is also caused by the later degradation of the peptides into bioactive peptides with antioxidative properties. Skim milk powder supplemented in the yogurt preparation step could be a source of antioxidants. In the skimming process, although various hydrophobic antioxidants such as carotenoids, retinol, and tocopherol and milk-soluble components were removed, other strong antioxidants such as thiols, ascorbate, and whey proteins were retained [28]. Ginseng is an excellent antioxidant supplier with various antioxidant ingredients. Different compositions of ginsenosides with polyphenols and flavonoids, such as gentisic acid, *p*- and *m*-coumaric acid, chlorogenic acid, and rutin that are known as representative ingredients of ginseng, possess powerful antioxidant activity [29]. In a previous study, the total polyphenolic and flavonoid content of soil-cultured ginseng and hydroponic ginseng was measured. While soil-cultured ginseng contained 17.36 mg gallic acid equivalent/g and 1.44 mg kaempferol/g, respectively, hydroponic ginseng had 31.80 mg gallic acid equivalent/g and 2.31 mg kaempferol/g, respectively [20]. The total phenolic and flavonoid content of hydroponic ginseng was roughly twofold higher than that of soil-cultured ginseng, and these results were consistent with the results of antioxidant analysis shown in Figure 1 and Figure 3. The antioxidant property of ginseng was shown to be exerted by the increased levels of superoxide dismutase and glutathione peroxidase-like enzymes in HepG2 hepatoma cells [30]. In a clinical study, the antioxidant effect of ginseng was reported as owing to the decreased activity of methane dicarboxylic aldehyde and reactive oxygen species levels in serum [31]. When free radicals such as superoxide (O^−^_2_) are produced at the tissue or cell level, they can induce the activation of macrophages and lipid peroxidation, which results in severe health dysfunction. The suppressive effect on oxidative damage by ginseng could have various health effects on aging, diabetes, hepatotoxicity, kidney diseases, vascular diseases, and neurological disorders [32,33]. Therefore, maximization of the functional properties of ginseng and its application to various types of food vehicles are required to improve the quality of life.

Many studies have noted that the addition of ginseng extract to other food products generally has negative sensory scores owing to the bitterness, despite the biofunctional activities of this ingredient toward human health [14,18,34]. Ginseng extract might contain triterpenoid peptides or propylene glycol, which may be responsible for the bitter taste of ginseng extract [35]. However, our study revealed that the addition of the ginseng extract at a concentration of 0.5% did not significantly affect the total quality score.

## 5. Conclusions

Ginseng has been widely used as a traditional medicinal plant owing to its various biological activities for health. However, there are many limitations and difficulties in cultivation depending on environmental conditions, including regional differences and long cultivation times. Hydroponic ginseng is being explored along with studies on its biological activity as an alternative ginseng source. Hydroponic ginseng has high commercial value because of its short cultivation period and abundant bioactive compounds. Our hypothesis was that yogurt supplemented with hydroponic ginseng extract would have higher antioxidant activity than the control because it has high antioxidant activity. The results of this study suggest that hydroponic ginseng can be practically applied to high-value dairy products to maintain human health without low consumer acceptability.

## Figures and Tables

**Figure 1 foods-10-00639-f001:**
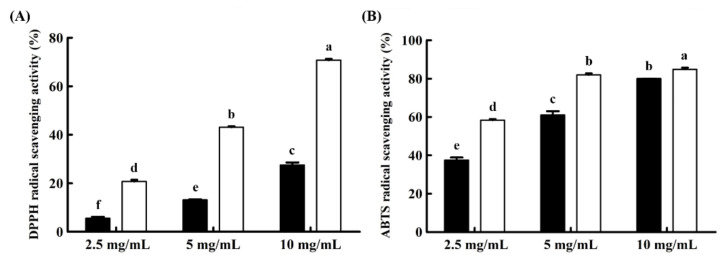
Antioxidant activity of ginseng extracts. Two radical scavenging assays were performed, (**A**) 2,2-diphenyl-1-picryl-hydrazyl (DPPH) and (**B**) 2-azinobis-(3-ethylbenzothiazoline-6-sulfonic acid) diammonium salt (ABTS). ■, soil-cultured ginseng extract; □, hydroponic ginseng extract. The graphs are expressed as mean value ± S.D. of triplicate experiments. Different letters represent significant differences between treated samples within one antioxidative analysis (*p* < 0.05).

**Figure 2 foods-10-00639-f002:**
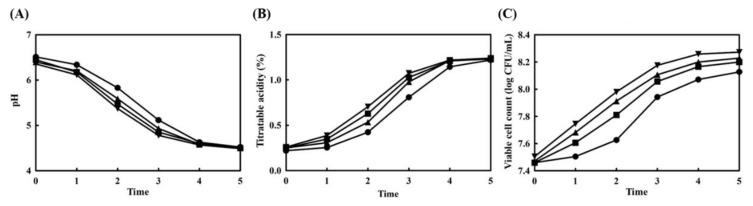
Change in pH (**A**), titratable acidity (**B**), and the viable cell count (**C**) of ginseng-fortified yogurts. ●, control yogurt without ginseng extract; ■, 1% soil-cultured ginseng extract-fortified yogurt; ▲, 0.5% hydroponic ginseng extract-fortified yogurt; ▼, 1.0% hydroponic ginseng extract-fortified yogurt.

**Figure 3 foods-10-00639-f003:**
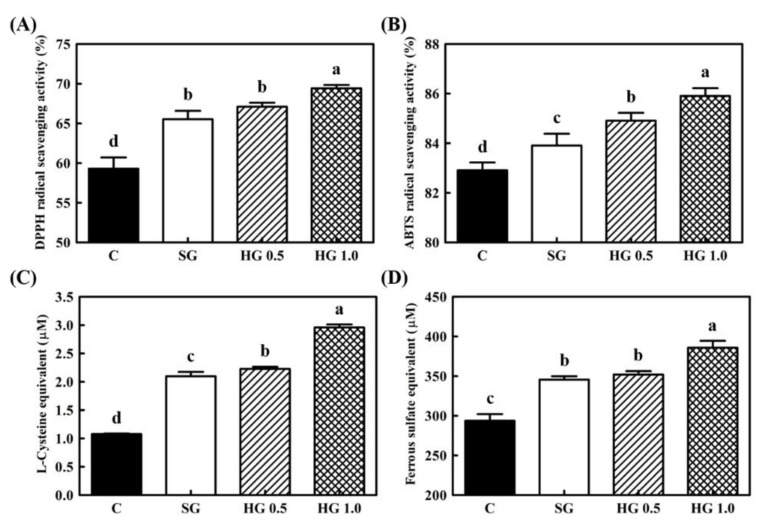
Antioxidant activity of ginseng extract-fortified yogurts. Four different antioxidant assays were performed: (**A**) 2,2-Diphenyl-1-picryl-hydrazyl (DPPH) radical scavenging assay, (**B**) 2-azinobis-(3-ethylbenzothiazoline-6-sulfonic acid) diammonium salt (ABTS) radical scavenging assay, (**C**) reducing power assay, and (**D**) ferric reducing antioxidant power FRAP assay. ■, C: Control yogurt without ginseng extract; □, SG: 1.0% soil-cultured ginseng extract-fortified yogurt; ▨, HG 0.5: 0.5% hydroponic ginseng extract-fortified yogurt; ▩, HG 1.0: 1.0% hydroponic ginseng extract-fortified yogurt. The bars represent mean ± standard deviation of at least triple repetitive analysis. Different letters on the error bars indicate a significant difference between sample groups (*p* < 0.05).

**Table 1 foods-10-00639-t001:** Physicochemical properties of ginseng extract-fortified yogurt.

Sample Type ^(1)^	Nutritional Composition (%)					Light Value ^(2)^		
	Fat	Protein	Lactose	Total Solid	Ash	L*	A*	B*
C	3.32 ± 0.15	4.15 ± 0.21 ^c^	7.28 ± 0.71	14.71 ± 0.27 ^c^	0.76 ± 0.02 ^b^	93.96 ± 0.26 ^a^	−2.92 ± 0.06 ^d^	5.91 ± 0.23 ^d^
SG	3.45 ± 0.10	4.03 ± 0.21 ^c^	7.30 ± 0.14	15.72 ± 0.20 ^b^	0.77 ± 0.01 ^b^	92.82 ± 0.32 ^b^	−2.67 ± 0.03 ^b^	8.62 ± 0.41 ^c^
HG 0.5	3.43 ± 0.16	4.72 ± 0.15 ^b^	7.33 ± 0.19	15.42 ± 0.12 ^b^	0.84 ± 0.01 ^a^	92.48 ± 0.26 ^b^	−2.82 ± 0.02 ^c^	9.07 ± 0.23 ^b^
HG 1.0	3.50 ± 0.14	5.30 ± 0.38 ^a^	7.33 ± 0.10	16.39 ± 0.28 ^a^	0.87 ± 0.03 ^a^	90.99 ± 0.73 ^c^	−2.30 ± 0.10 ^a^	11.11 ± 0.33 ^a^

^a–d^ Values are represented as mean ± standard deviation of at least triplicate experiments. Different superscripts for the identical parameters indicate a significant difference (*p* < 0.05) analyzed using ANOVA. ^(1)^ C: Control yogurt; SG: 1.0% soil-cultured ginseng extract-fortified yogurt; HG 0.5: 0.5% hydroponic ginseng extract-fortified yogurt; HG 1.0: 1.0% hydroponic ginseng extract-fortified yogurt. ^(2)^ L*: Range from 0 (black) to 100 (white) implying lightness; A*: Range from green (negative value) to red (positive value) color; B*: Range from blue (negative value) to yellow (positive value) color.

**Table 2 foods-10-00639-t002:** Sensory evaluation of yogurt supplemented with various ginseng extracts.

Items	Hydroponic Ginseng Extract Concentration Added			
	Control	0.1%	0.5%	1.0%
Color	5.58 ± 2.89 ^b,(1)^	5.96 ± 1.40 ^a,b^	6.03 ± 0.89 ^a^	4.88 ± 0.95 ^c^
Texture	4.79 ± 1.17 ^a^	4.68 ± 1.27 ^a,b^	4.65 ± 1.23 ^a,b^	4.27 ± 1.19 ^b^
Flavor	5.04 ± 1.11 ^a^	4.93 ± 0.95 ^a,b^	4.92 ± 1.10 ^a,b^	4.50 ± 1.03 ^b^
Taste	5.38 ± 1.17 ^a^	5.04 ± 1.61 ^b^	4.96 ± 1.04 ^b^	3.88 ± 1.21 ^c^
Overall acceptance	5.54 ± 0.99 ^a^	4.15 ± 1.05 ^b^	5.04 ± 1.04 ^a^	3.54 ± 1.33 ^c^

^(1)^ Mean ± S.E. Superscript lowercase letters for identical parameters indicate a difference using the Duncan’s test (*p* < 0.05).

## Data Availability

Data sharing not applicable.

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
