# Peer review of "Antioxidant Effect and Sensory Evaluation of Yogurt Supplemented with Hydroponic Ginseng Root Extract"

_foods, 2021, doi:10.3390/foods10030639_

Round 1

Reviewer 1 Report

The paper concerns an up-to-date issue of  plant origin  additives with high antioxidant potential which are used in the production of functional dairy products. This kind of  supplementation is nowadays often described and many authors analyze the influence of different plant extracts on the properties of the product and on its bioactivity. Ginseng root has been used as a traditional medicinal plant in Asian countries, but nowadays its potential is appreciated worldwide. In my opinion  the originality of research comes mainly from the different method of cultivation of the plant, much shorter than the traditional soil cultivation, which, at the end, gives higher antioxidant properties of its extract, probably due to toe higher content of some antioxidant compounds. It is difficult to judge as the Authors did not provide any characteristics of the ginseng extract obtained from the hydroponic cultures for example composition, main groups of compounds responsible for the antioxidant activity: ginsenoside, phenolic and flavonoid  compounds. As the authors pointed (line 262-266): “Hydroponic systems have advantages not only for facilitating productivity but also for controlling environmental factors. This system enables the acceleration of growth and increases the functional compounds of ginseng through the regulation of nutritional supply, light intensity, and temperature” – this way of cultivation may also influence the composition of functional compounds responsible for antioxidant activity.

Authors of the research used a simple research scheme, all the results are clearly presented. The context of the paper fits within the wide stream of functional food topic, in my opinion the research might be valuable although there are quite a lot of similar papers. Some information had to be provided or clarified.

I would appreciate the authors comments on:

  1. How authors explain so high total solids level in analyzed yoghurts: 29,42% to 32,92% (in Table 1)? How is it possible, as in Materials and methods they stated that:

“Milk was supplemented with skim milk powder to yield 11% in solid matter (line 93)”.

Please provide the detailed information what kind of milk was used and how much of skim milk powder was added in yoghurt production and how yoghurts were produced?  The total solids at the level of about 30% are characteristic for fresh cheeses, not for yoghurts, so in that context the presented values seem unreasonable. Even if we assume that for production sheep milk was used, which is characterized by higher total solids (aprox. 17%) it is still lower that the presented values and is not in line with the information in line 93.

What’s more, the levels of components making up the total solids in Table 1 (fat, protein, lactose, ash) give after addition 15,51% (for control), 15,55% for SG; 16.32% for HG 0,5 and 17 fort HG 1,0; but not  29,42% to 32,92%. What is the author comment on that?

  1. Why the analysis was performed only for 5h during yoghurt production? It is rather a short time especially that yoghurts usually have at least about 7-14 days of expiry date, and rarely are consumed just after production. It would be more informative if anything has been said about the stability of the antioxidant activity during yoghurt storage. Was it due to the low long-term antioxidant stability of the extract?  I would appreciate any comment on that.
  2. Why the media used for total number of viable cells count determination was bromocresol purple medium (MB Cell). The medium used by the authors gives no information about growth of which of groups of the microbes used as starter culture (Lactobacillus acidophilus, Bifidobacterium longum, and Streptococcus thermophilus) was stimulated by the ginseng extract. Usually MRS agar is used for Lactobacilli and M17 for Sterptococci. Bifidobacteria grow in milk rather poor so I wouldn’t expect any growth of that group during 5h of production.
  3. On Fig 2 B and C, there is no units for titrable acidity nor for the total number of viable cells count. Was it % of lactic acid and log10 of cfu respectivly?? It need to be clarified and corrected on Fig 2 B and C. Especially in the case of total number of viable cells, as yoghurts had to be characterized by the level of 106 -107 cfu of viable cells at the last day of the expiry date, it is important to determine the starting level of yoghurt microflora.
  4. Lines 196 -200: “Protein, total solid, and ash content of ginseng extract-supplemented yogurt showed a dose-dependent increasing pattern with significant differences. Although the concentration of HG 0.5 was 2-fold lower than that of SG, it contained slightly more protein and ash. Total solid content also showed a dose-dependent increasing pattern. However, the fat and lactose content did not show significant differences between all types of yogurt samples” –

- it is difficult to discuss the correlation as no characteristics of ginseng extract (even just basic: protein/carbohydrates/fat ) was provided.

Depending on the Authors answers to comments above the manuscript section should be corrected accordingly in the proper sections.

Author Response

I sent a response letter with an attached file.

Reviewer 2 Report

The manuscript is well-organised and readable. I have some suggestions:

1) Figure 2 and Figure 3 have the same number 

2) Figure 1 and Figure 2 (3 in fact) -> Authors have written about significant differences marked by different letters, but I don't see the letters on the charts. 

3) ● ■ ▲ ▼ on the Figure 2 should be larger. It is difficult to see differences between lines on the charts. 

4)Lines 201-208 --> Were the lyophilised SG and HG extracts differ in color or similar? Are differences in yogurt colour due to differences in the extracts color or it occurs after the addition of extracts to yogurt?

5) I think, that composition of the extracts should be included in the manuscript. Authors write about differences in the composition (line 315-325) but this differences are for methanolic extracts obtained by another methods.

Author Response

(The authors gave the same response as above.)

Round 2

Reviewer 1 Report

I accept all authors answers to my comments and all corrections done in the text.  I'm glad I could help to correct the obvious mistake (dry matter level). 

As to Your answer about the short period after which yoghurts were analysed: (5h during yoghurt production) -

“we tried to consume yogurt rapidly before the distinctive flavor of ginseng was disappeared for exact evaluation of acceptance. Additionally, we added this information on manuscript as your comment”

It still suggests that the extract may not be stable during  storage. I would advise to analyze its stability during longer storage (7-10 day) as well as the antioxidant activity in the future research.